# Prediction of Composite Mechanical Properties: Integration of Deep Neural Network Methods and Finite Element Analysis

**Kimia Gholami [1,*], Faraz Ege [2] and Ramin Barzegar [3]**

[1] Department of Computer Science, Florida Institute of Technology, Melbourne, FL 32901, USA
[2] Department of Mechanical and Civil Engineering, Florida Institute of Technology, Melbourne, FL 32901, USA
[3] School of Engineering, Atilim University, Ankara 06830, Turkey
[*] Correspondence: fgholami2021@my.fit.edu

**Abstract:** Extracting the mechanical properties of a composite hydrogel; e.g., bioglass (BG)–collagen (COL), is often difficult due to the complexity of the experimental procedure. BGs could be embedded in the COL and thereby improve the mechanical properties of COL for bone tissue engineering applications. This paper proposed a deep-learning-based approach to extract the mechanical properties of a composite hydrogel directly from the microstructural images. Four datasets of various shapes of BGs (9000 2D images) generated by a finite element analysis showed that the deep neural network (DNN) model could efficiently predict the mechanical properties of the composite hydrogel, including the Young's modulus and Poisson's ratio. ResNet and AlexNet architecture were tuned to ensure the excellent performance and high accuracy of the proposed methods with R-values greater than 0.99 and a mean absolute error of the prediction of less than 7%. The results for the full dataset revealed that AlexNet had a better performance than ResNet in predicting the elastic material properties of BGs-COL with R-values of 0.99 and 0.97 compared to 0.97 and 0.96 for the Young's modulus and Poisson's ratio, respectively. This work provided bridging methods to combine a finite element analysis and a DNN for applications in diverse fields such as tissue engineering, materials science, and medical engineering.

**Keywords:** composite hydrogel; tissue engineering; deep learning

## 1. Introduction

Almost all natural and artificial materials exist in the form of composites. Extracting the material properties of composites is essential in many applications; for instance, marine construction and buildings [1–3], materials science, and medical and tissue engineering [4]. In tissue engineering, biocompatible materials (i.e., biomaterials) are utilized to generate hydrogels or other scaffolds to repair or replace damaged or diseased tissues [5]. The selection of biomimetic materials for hosteler tissues is based on the physicochemical properties of the native tissue to reduce the chance of scar tissue building at the interphase. To improve the mechanical properties of COL, which are one of the most abundant proteins in mammals [6], BGs can be used to formulate composite scaffolds [7]. Many papers showed that the incorporation of BGs into COL could significantly improve the mechanical properties of COL such as increased stiffness [5,8], reduced swelling, improved stability and rheological properties (i.e., yield stress) [9], etc. In addition, the mechanical properties of BGs-COL depend on the BGs' concentration, spatial distribution, particle size, and fabrication process [10]. For instance, Wang et al. [11] showed that crosslinking of BGs-COL can affect composite hydrogels' mechanical properties (crosslinked BGs-COL had a 37 kPa Young's modulus compared to a 5 kPa Young's modulus for non-crosslinked BGs-COL). These mechanical properties can primarily be determined experimentally, analytically [12], or computationally [13] or by using finite element methods (FEM) [14–18].

Theoretical homogenization methods such as the double-inclusion method, which consists of an ellipsoidal inclusion that contains an ellipsoidal heterogeneity and is embedded in an infinitely extended homogeneous domain [19]; self-consistent approaches; and the Mori–Tanaka mean field method generally can solve simple microstructures and are impracticable for complex structures [19–22]. For a heterogeneous material, the representative volume element (RVE) technique can extract the effective mechanical properties of the composite material in the FEM [23–25]. RVE can be considered a volume that effectively includes sampling all microstructural heterogeneities in the composite. It must remain small enough to be considered a volume element of continuum mechanics [26]. Several boundary conditions can be prescribed to impose a given mean strain or stress on the material element. Omairay et al. [25] developed a plugin that can automatically apply the concept of the periodic RVE homogenization and periodic boundary conditions in the Abaqus software to estimate the homogenized effective elastic properties of the composite. Kim et al. [27] developed a Python script in the Abaqus user interface to generate simulation models of the RVEs that consisted of inclusions and a matrix. In addition, they used a random sequential expansion (RSE) algorithm to create a dataset of circular shape inclusions in the matrix. Ye et al. [15] generated many RVEs with various types of complex structures that consisted of arbitrary (either regular or irregular) shapes of inclusions. These papers considered either inclusion as circular or elliptical or arbitrary shapes. This study assessed four datasets of uniform shapes, non-uniform shapes, irregular shapes, and a combination of the three mentioned datasets as a full dataset. Despite the fact that applying RVEs in the FEM can be an easy-to-use tool to extract the mechanical properties of composite hydrogels, this method also requires high computational costs and is time-consuming. To tackle this challenge and speed up this process, machine learning (ML) and deep learning (DL) technologies were introduced in the application of materials science and composite materials design.

Recently, ML and DL algorithms in artificial intelligence have become essential tools in vast applications [28,29], especially in composite material design and materials engineering, which rely on their power to predict different material properties such as the mechanical properties [15,27,30–35], thermal conductivity, and thermal expansion coefficients [36]. Due to the structural complexity of composites and novel materials, optimizing and predicting materials' behaviors can affect enormous spaces of design untraceable by conventional methods. By tackling this challenge with a sufficient dataset that is properly preprocessed, ML aims to learn the mapping between the input and expected output by using the high-dimensional feature vector from the original data. Recent advances in ML methods have resulted in many prospects for overcoming traditional ways of designing composites, predicting material properties, and optimizing composite structures [15,27,35,37]. For instance, Shokrollahi et al. [5] used a finite-element-based ML model to predict the mechanical properties of composite hydrogels with circular shapes of BGs distributed in the COL. They concluded that ML could effectively predict the mechanical properties of the composite hydrogels, including the Young's modulus and Poisson's ratio, which showed R-values of 95% and 83%, respectively. Bhaduri et al. [38] integrated 25 FEM analyses of 10 fiber composites with U-Net architecture to predict the local stress field in a fiber-reinforced matrix. Yang et al. [32] predicted the entire stress–strain behavior of binary composites using convolutional neural networks (CNN) and a principal component analysis (PCA). Li et al. [39] developed a genetic algorithm optimized back propagation (GABP) neural network model to predict the transverse mechanical properties of unidirectional carbon-fiber-reinforced polymer (CFRP) composites with microvoids. Zhang et al. [40] predicted the mechanical properties of a composite laminate, including the failure factor of Puck theory. However, ML models have not been applied to predict the mechanical properties of BGs-COL with regular and irregular BGs (microstructure images of BGs-COL show that BGs do not always have circular shapes [11]). This study used larger datasets of BGs-COL microstructure images compared to a previous study [5] and various shapes of BGs in the COL to predict the mechanical properties of BGs-COL by bridging FEM and DL methods.

The purpose of developing this framework was to speed up the process of extracting the material properties of composite hydrogels compared to the traditional methods, which are mostly time-consuming and computationally expensive. Moreover, this study can serve as a surrogate model for predicting the Young's modulus and Poisson's ratio of composite hydrogels.

In this work, two well-known ML networks (ResNet [41] and AlexNet [42]) were trained and tuned on the four datasets of BGs-COL microstructure images to predict the Young's modulus and Poisson's ratio. A total of 9000 microstructural images of BGs-COL were generated in three categories: circular BGs shapes with the same diameter as a uniform dataset, BGs distributed in the COL with circular shapes and different diameters as a non-uniform dataset, and free-shape BGs embedded in the COL as an irregular dataset. In addition, the combination of the three generated datasets was considered the full dataset. These datasets were generated by a Python script and run in the FE simulation to extract their corresponding mechanical properties. The performances of the DNN networks were evaluated in terms of the R-value, MAE, and mean squared error (MSE), which demonstrated that the trained DNN models could perfectly predict the mechanical properties of BGs-COL and overcome the challenge of prediction using the traditional homogenization methods, thereby showing an excellent performance.

## 2. Materials and Methods

The workflow of the proposed framework that bridged an FE analysis with the CNN networks to predict the composite material's properties is illustrated in Figure 1. Firstly, three datasets (uniform, non-uniform, and irregular shapes of microstructural images of BGs-COL) were generated with 9000 microstructural images. These 200 × 200 pixel images were then automatically imported into the FE software, and their corresponding Young's modulus and Poisson's ratio were computed and extracted using a Python script. Two well-known DNN networks—ResNet and AlexNet—were implemented and trained on the datasets. Finally, tuned networks predicted the mechanical properties of the BGs-COL.

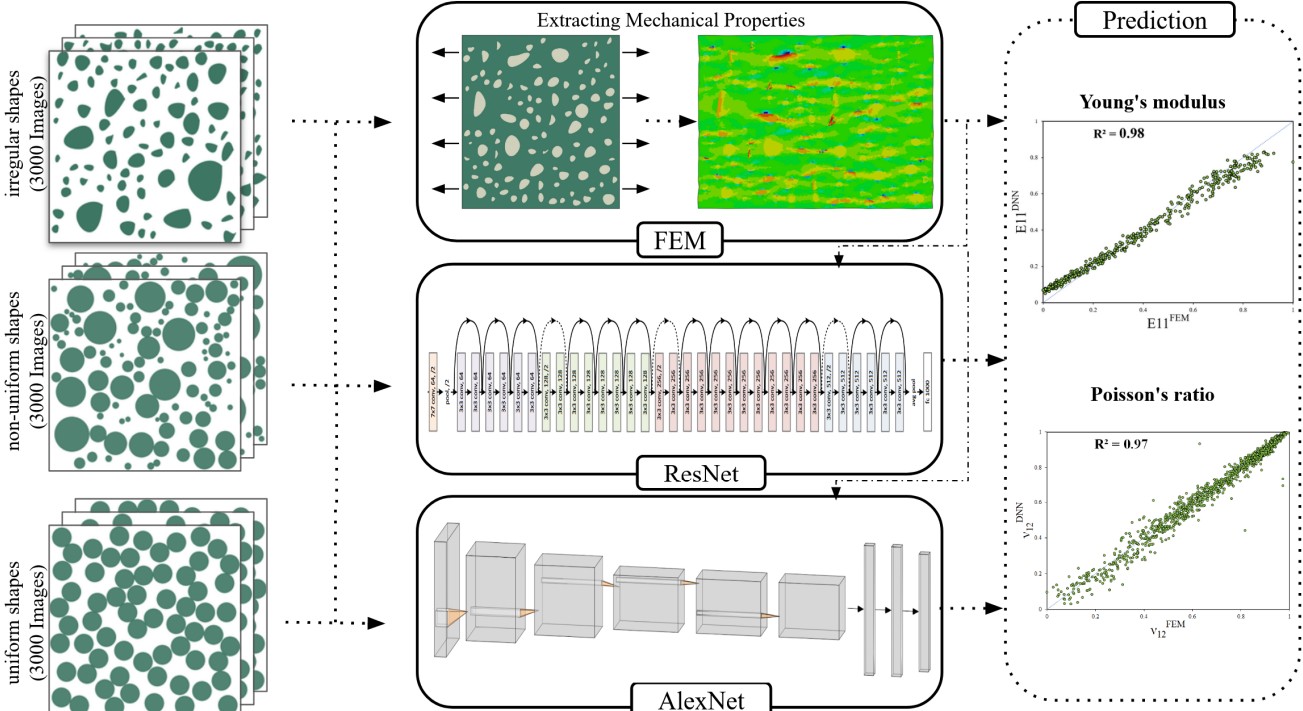

**Figure 1.** Overall framework that bridged FE analysis with DNN networks to predict the Young's modulus and Poisson's ratio of BGs-COL.

### 2.1. Simulation-Based Datasets

The BGs' shapes were divided into three categories and distributed in the COL using Python scripts as shown in Figure 2. To generate COL containing BGs of arbitrary numbers, sizes, and three different shapes, Python scripts were used. Firstly, a matrix size for COL of $20 \times 20$ μm was defined. For the regular shapes of BGs, a random number between 1 to 130 that represented the number of the BGs was picked. The diameter of the BGs was 1 μm for the uniform dataset and a range of 0.2 to 1.5 μm for the non-uniform dataset. Following that, the developed Python code picked two random numbers in the COL space (between 0 and 20) as the center of the first circle in the x and y directions, and the first BG was placed in the COL. For the next circles, the distance of the picked numbers as the new circle's centers was checked and had to be greater than the diameter of the neighbor's circles to avoid overlap between circles. This step was repeated for the remaining circles to generate the first BGs-COL image (Figure 2a,b). The following algorithm was employed to create irregular shapes of BGs as shown in Figure 2c. Firstly, the non-uniform code was used as previously described. Then, random vertices were generated in each circle's local polar coordinate system. The spline algorithm connected the vertices and generated irregular shapes of BGs in the COL. It must be noted that although BGs embedded in the COL were considered perfectly bonded for simplicity, the developed Python code could quickly be expanded to generate interphase between the BGs and COL.

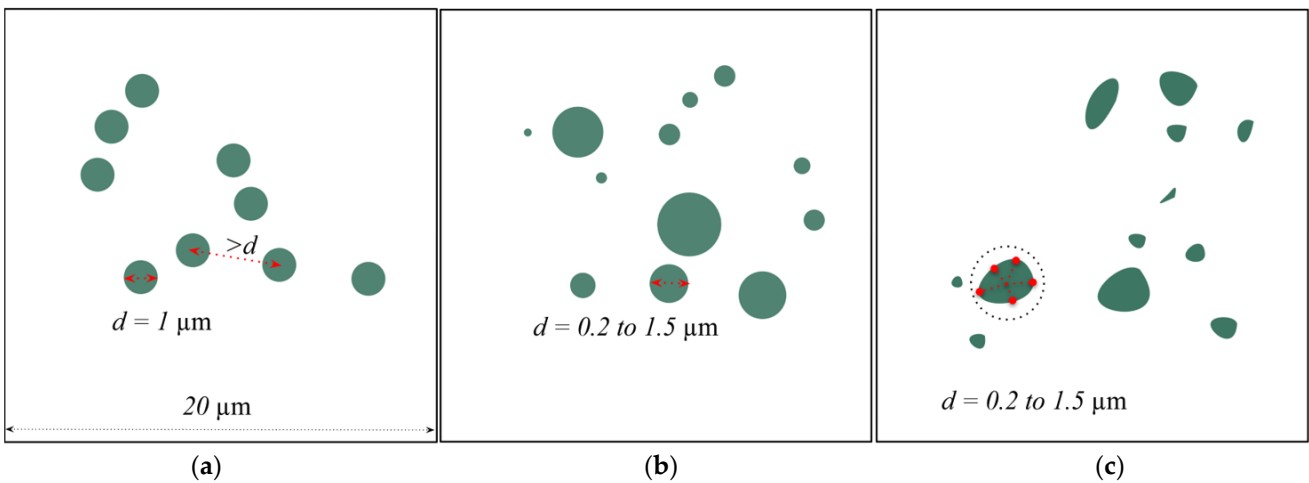

**Figure 2.** Three patterns of BGs embedded in the COL: (**a**) circular BGs shapes with the same diameter as a uniform dataset; (**b**) BGs distributed in the COL with circular shapes and different diameters as a non-uniform dataset; (**c**) free-shape BGs embedded in the COL as an irregular dataset.

The Python script, which was developed in the Abaqus/Explicit software version 2021, was motivated by [5,25]. This Python script was divided into two main phases to calculate the homogenized elastic properties by implementing the unified periodic RVE homogenization method concepts. Firstly, the code was used to generate the geometries and assigned material properties (Young's modulus (E) and Poisson's ratio (ν)) of the BGs as E = 76.7 GPa and ν = 0.261 [43] and COL as E = 3 kPa and ν = 0.49 [5]. The approximate global meshing size was chosen as 0.2 μm with two-dimensional generalized plane stress elements (CPS4R). To apply a uniform displacement of 20% of the COL length (4 μm), the code determined the boundary surfaces and RVE dimensions; then, by building nodal sets, the boundary conditions and displacement could be applied and the FE analysis could be conducted. The second phase was post-processing and calculating the Young's modulus and Poisson's ratio of the BGs-COL hydrogel. This step computed the first nodal forces at the affected boundary nodes and divided them by the affected surface area to provide the stress value. So, by dividing the calculated stress value by the applied axial strain of 0.20, the Young's modulus of the BGs-COL could be obtained.

Moreover, the transverse strain was simply the ratio of the change in height to the original height. The Poisson's ratio was estimated as the ratio of the transverse strain to the applied axial strain of 0.20. Detailed explanations can be found elsewhere [25]. The verification of the simulation results was conducted using [12,25].

*2.2. Machine Learning Approach*

ResNet and AlexNet architectures were implemented and trained to extract the elastic mechanical properties of the BGs-COL, including the Young's modulus and Poisson's ratio. The motivation for choosing these networks was the study by Ye et al. [30], which showed that ResNet had a higher accuracy in extracting the Young's modulus and Poisson's ratio of composite materials than DensNet [44]. ResNet was proposed by He et al. [41] while attempting to design a network that solved the vanishing gradient problem. ResNet was developed with many layers: 34, 50, 101, 152, and even 1202. The version of ResNet used in this paper had 34 layers (ResNet34). The basic block diagram used in the ResNet architecture is called ResBlocks. In ResNet, a technique called skip connections was used to connect activations of a layer to further layers by skipping some layers in between. In addition, the AlexNet network also was trained because it has shown a significant breakthrough in ML and computer vision, especially in image classification. It is a very large network with over 6 million parameters. AlexNet consists of five convolutional layers and three fully connected layers. In the last fully connected layer of both networks, the rectified linear unit (ReLU) was chosen [45] as the activation function for the regression problems. The ReLU function is denoted in Equation (1):

$$f(z) = \max(0, z) \tag{1}$$

Based on the ReLU function, there is a drawback when most of the inputs to ReLU activation are in the negative range; subsequently, ReLU returns zero as outputs, a large part of the network becomes inactive, and it is unable to learn further. For this reason, He et al. [46] modified the ReLU function and called it the Leaky ReLU function, which has a nonzero but slight slope of 0.01 when $z < 0$ as denoted in Equation (2):

$$f(z) = \begin{cases} 0.01z \; for \; z < 0 \\ z \; for \; z \geq 0 \end{cases} \tag{2}$$

The performance also was compared with the Leaky ReLU activation function. Max–min normalization methods were chosen to scale all target inputs to the same order of magnitude [47]. In the max–min normalization methods, the data were scaled to the range of [0, 1]. This method converted the input value *x* of the attribute *X* to *xnorm* the range [*low*, *high*] by using the formula in Equation (3):

$$x_{norm} = \frac{(high - low) \times (x - minX)}{maxX - minX} \tag{3}$$

where *minX* and *maxX* are the minimum and maximum values of the attribute *X* of the input dataset. MSE was used as the loss function in the DNN networks. These DNN architectures were implemented by using TensorFlow and Keras in the Jupyter Notebook IDE [48]. The ML models were run on a computer equipped with a Ryzen 7 5800X processor, 64 GB of DDR4/2666 MHz memory, and an Nvidia GeForce RTX 3070 GPU.

## 3. Results

In the present study, three datasets were trained: uniform, non-uniform, and irregular shapes. In addition, all datasets were combined and trained together as a full dataset. The training and testing datasets contained 2550 and 450 images, respectively (using an 85:15 train–test split). The full dataset also contained 9000 images (7650 and 1350 for training and testing, respectively). All networks were trained in 20 repeated epochs. Additional statistical descriptors such as the MAE, MSE, R-value, and average were provided to

evaluate the performance of our DNN networks. Figures 3 and 4 depict the Young's modulus and Poisson's ratio predicted by ResNet and AlexNet, respectively, plotted against the ground truth that the FEM generated. The centerline y = x shows that both trained ResNet and AlexNet networks could perfectly learn and map the microstructure images to the mechanical properties; e.g., Figure 3a–c,g indicate that the Young's modulus could be predicted by ResNet with a more than 0.96 R-value in all datasets. In addition, an efficient metric of model performance for the material descriptor was the MAE, which provides confidence estimates of the model's prediction. When comparing the MAE to the range of values (the range of values was 1 because the max–min normalization method was used), the MAE was relatively small. For the Young's modulus predicted by ResNet, the MAE compared to the range was less than 8% for all datasets (Table 1). The Poisson's ratio extracted from the ResNet model trained with four datasets is plotted against the ground truth (FEM) in Figure 3d–f,h. It can be seen that ResNet could perfectly predict the Poisson's ratio with more than a 0.94 R-value for the uniform, non-uniform, and full datasets; while the irregular datasets showed a lower R-value of 0.85. In addition, the MAE of the Poisson's ratio extracted by ResNet compared to the range was less than 8% for all datasets.

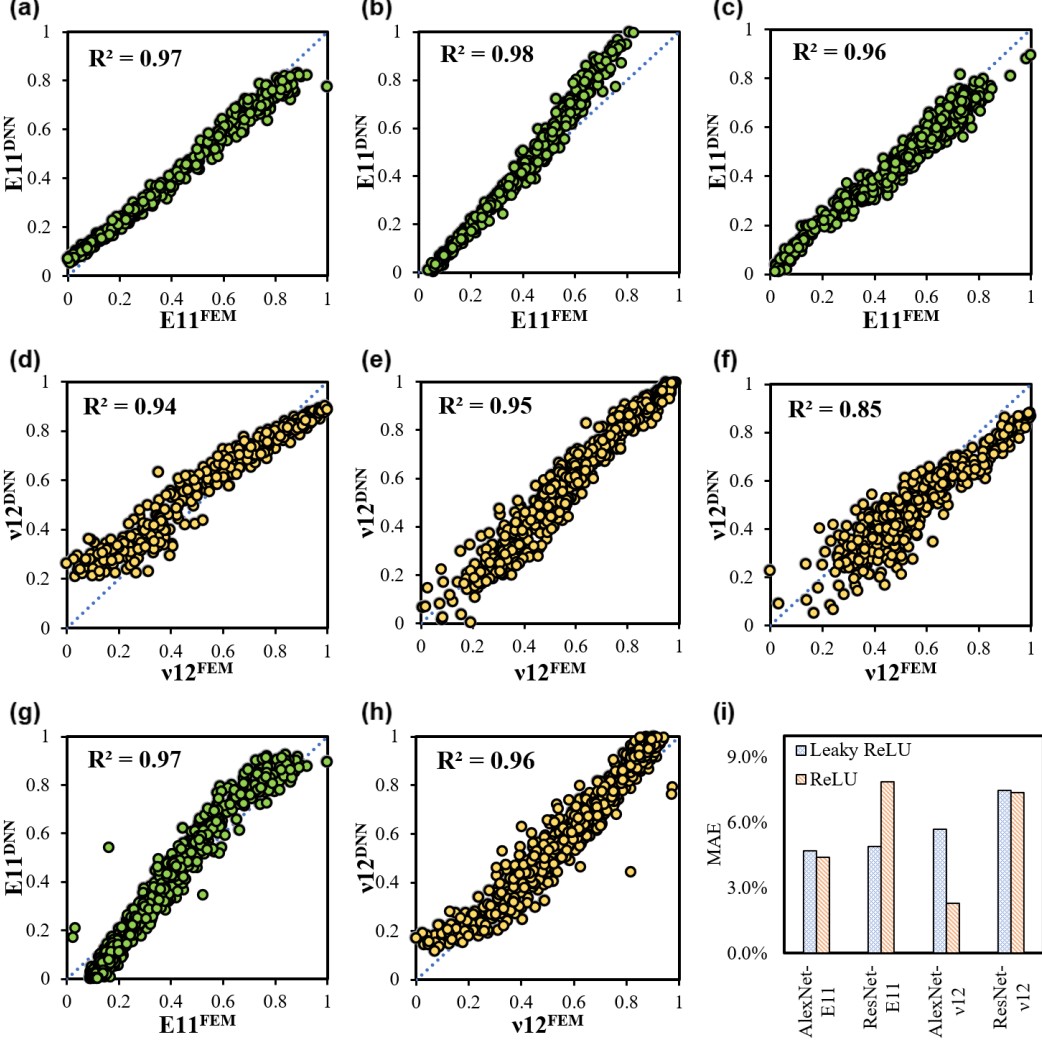

**Figure 3.** Performance of the ResNet model in extracting the elastic property of the composite hydrogel including the Young's modulus (E11) and Poisson's ratio (ν12) on the (**a**,**d**) uniform dataset, (**b**,**e**) non-uniform dataset, (**c**,**f**) irregular dataset, and (**g**,**h**) full dataset. (**i**) Comparison of the ReLU and Leaky ReLU activation function performance in terms of MAE for ResNet and AlexNet on the full dataset.

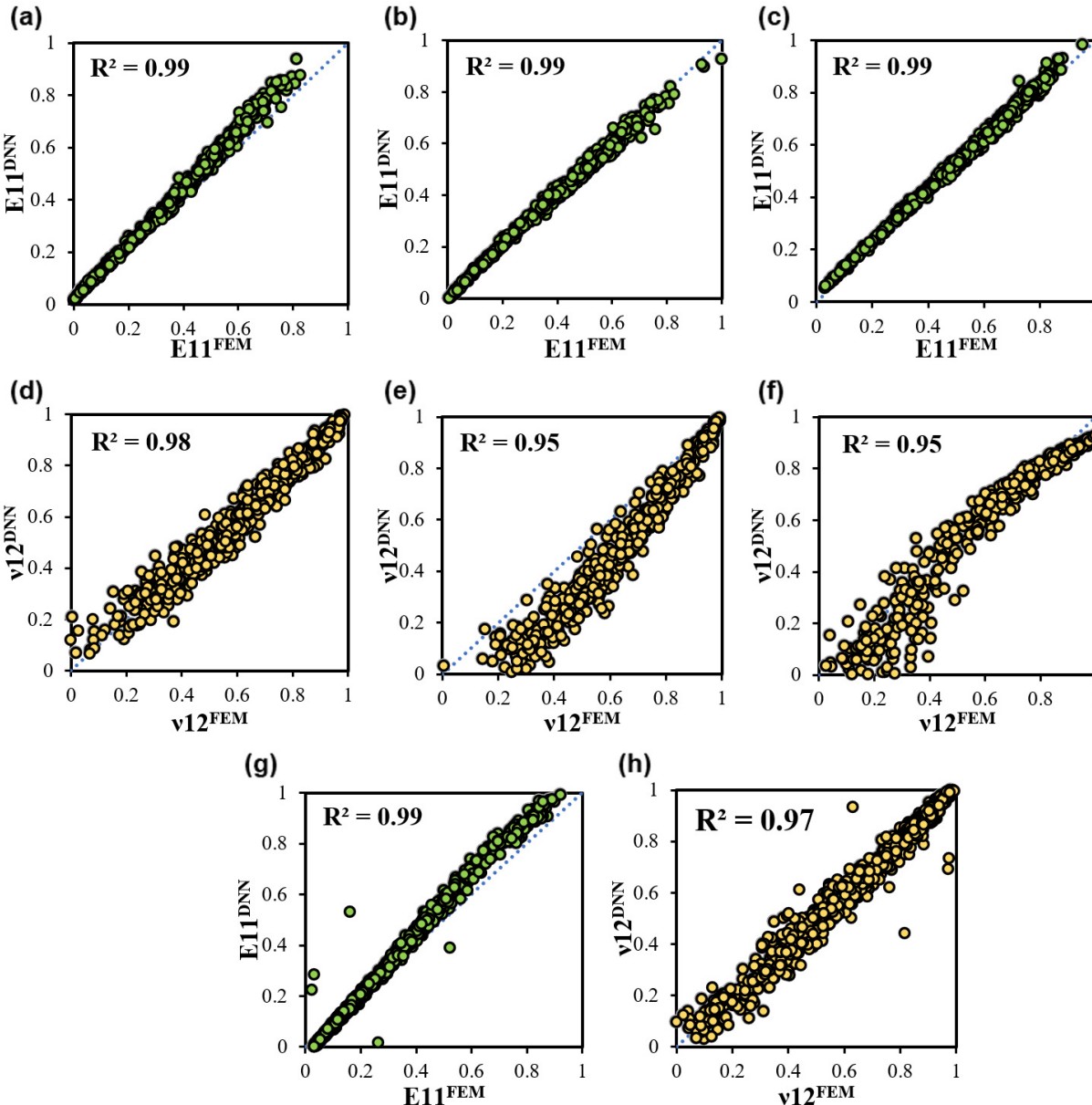

**Figure 4.** Performance of the AlexNet model in extracting the elastic property of the composite hydrogel including the Young's modulus (E11) and Poisson's ratio (v12) on the (**a**,**d**) uniform dataset, (**b**,**e**) non-uniform dataset, (**c**,**f**) irregular dataset, and (**g**,**h**) full dataset.

Comparing all provided metrics of the AlexNet network showed a better performance than ResNet in predicting the Young's modulus and Poisson's ratio except for the MAE of the Poisson's ratio in the non-uniform dataset. Relatively tight confidence intervals on the errors indicated that the ResNet and AlexNet model architectures perfectly mapped the microstructural images of BGs-COL to the elastic mechanical properties. ReLU and Leaky ReLU were chosen as the activation functions in the last fully connected layer. Subsequently, the performances of our DNN models were compared using the MAE of the networks on the full dataset. In the AlexNet model, ReLU had a better performance than Leaky ReLU (a 0.3% and 3.4% lower MAE for the Young's modulus and Poisson's ratio, respectively).

**Table 1.** Using AlexNet and ResNet, we evaluated the performance of our implemented ResNet and AlexNet networks with an 85:15 train–test split. The performances were measured according to the error between the FEM results and the predicted Young's modulus and Poisson's ratio derived from the trained AlexNet and ResNet. Statistical descriptors are provided for context (MAE, MSE, R-value, and average of distributed data using the max–min normalization method).

| Network | Dataset | Properties | MAE | MSE | $R^2$ | Average |
|---|---|---|---|---|---|---|
| AlexNet | Uniform | E11 | 0.032 | 0.001 | 0.99 | 0.373 |
| | | $\nu$12 | 0.053 | 0.004 | 0.98 | 0.587 |
| | Non-uniform | E11 | 0.011 | 0.001 | 0.99 | 0.382 |
| | | $\nu$12 | 0.143 | 0.026 | 0.95 | 0.591 |
| | Irregular | E11 | 0.013 | 0.001 | 0.99 | 0.491 |
| | | $\nu$12 | 0.068 | 0.006 | 0.92 | 0.561 |
| | Full | E11 | 0.044 | 0.003 | 0.99 | 0.258 |
| | | $\nu$12 | 0.023 | 0.001 | 0.97 | 0.716 |
| ResNet | Uniform | E11 | 0.053 | 0.005 | 0.97 | 0.373 |
| | | $\nu$12 | 0.062 | 0.007 | 0.94 | 0.587 |
| | Non-uniform | E11 | 0.056 | 0.005 | 0.98 | 0.382 |
| | | $\nu$12 | 0.046 | 0.003 | 0.95 | 0.591 |
| | Irregular | E11 | 0.045 | 0.003 | 0.96 | 0.491 |
| | | $\nu$12 | 0.077 | 0.008 | 0.85 | 0.561 |
| | Full | E11 | 0.079 | 0.008 | 0.97 | 0.258 |
| | | $\nu$12 | 0.074 | 0.007 | 0.96 | 0.716 |

Similarly, in the ResNet model, the predicted Poisson's ratio had a 0.1% lower MAE with the ReLU activation function. However, the Young's modulus predicted by ResNet showed that Leaky ReLU could perform better than ReLU (3.0% lower MAE). As a result, ReLU was selected for all networks and datasets depicted in Figure 3i.

## 4. Discussion

This paper proposed a deep-learning-based approach to extract the mechanical properties of a composite hydrogel directly from the microstructural images. By integrating FEM and a Python script, three datasets of microstructural images were generated with uniform, non-uniform, and irregular shapes of BGs that contained 3000 2D images per each dataset and their corresponding Young's moduli and Poisson's ratios. Combining all datasets also provided a larger dataset containing 9000 images. Then, ResNet and AlexNet were implemented and trained on the datasets. The performance of ResNet and AlexNet networks were measured with an 85:15 train–test split by calculating the errors between the FEM and DNN results. Statistical descriptors that included the MAE and R-value showed that both networks had a great performance in predicting the Young's modulus and Poisson's ratio, although AlexNet had a much better performance. This work could guide the design of BGs-COL and other composite hydrogels and provide a framework for clinicians to predict composite hydrogels' mechanical properties rapidly.

This study fed 2D images to ResNet and AlexNet networks and mapped them to the corresponding Young's modulus and Poisson's ratio extracted via FEM. In the first dataset, uniform, arbitrary numbers of BGs with a diameter of 1 μm were embedded in the COL, and 3000 microstructural images were generated. This dataset had the simplest BG geometries and the best performance when AlexNet was trained (0.99 and 0.98 R-values for the Young's modulus and Poisson's ratio, respectively). The average normalized Young's modulus and Poisson's ratio in the uniform dataset were 0.372 and 0.587, respectively. These metrics showed that for most of the images, the Young's modulus had small values with some larger values. However, the average of the normalized Poisson's ratio indicated an equal distribution in the range of 0 to 1. The main reason for this distribution, which was repeated in all datasets, can be referred to as the volume fraction of BGs in the COL, which had a linear relation with the Young's modulus and Poisson's ratio. Shokrollahi et al. [5] showed that the Young's modulus would increase when increasing the volume fraction

of BGs in the COL. Contrarily, the Poisson's ratio would decrease with a greater volume of BGs in the COL. Referring to this conclusion, in the uniform dataset, most images had a lower volume fraction of BGs in the COL. A Python script was also implemented to generate irregular shapes of BGs in the COL in order to have a variety of datasets and to ensure that the datasets covered any possible BG shapes compared to real composites.

Our main motivation was to address a gap in the integration of FEM and robust DNN methods to predict the mechanical properties of a composite hydrogel [34,49,50], and we were inspired by the research in [5,30]. Ye et al. [30] concluded that ResNet34 and ReLU had a better performance in predicting a composite's mechanical properties than DensNet and Leaky ReLU. Similarly, ReLU had a better performance; however, the results showed that AlexNet was more accurate than ResNet34. When comparing the developed non-uniform dataset results containing 3000 images with the research of Shokrollahi et al. [5], they had a smaller dataset containing 2000 images with the same shapes and showed that better R-values for both the Young's modulus and Poisson's ratio were obtained. To achieve these DNN performances, preprocessing of the data and tuning of the hyperparameters were conducted in both ResNet and AlexNet networks. First, a max–min normalization method was applied, which could help to improve DNN networks' performance [47] and the networks to be unbiased. In addition, two activation functions were used in the last fully connected layer (with a better performance of ReLU than Leaky ReLU), and MSE was chosen as a loss function for both ResNet and AlexNet. The rest of the hyperparameters of ResNet and AlexNet remained as the default settings. When comparing the performances on all datasets, it was seen that the DNN networks performed well and were not shape-oriented. Whether the BGs' geometry was circles or irregular shapes, the DNN extracted important features and mapped them to the corresponding outputs.

The dataset used in this study contained 2D images with different shapes and random distributions of BGs in the COL. However, the proposed method could be expanded to predict the three-dimensional mechanical properties of BGs-COL. For simplicity, it was considered that the BGs bonded perfectly in the COL; the Python code could be easily modified to generate an interphase between the BGs and COL. In addition, the Young's modulus and Poisson's ratio were extracted; this framework could extract different material properties such as stress–strain curves and distributions, etc.

## 5. Conclusions

This study showed that by bridging DNN with FEM, the mechanical properties of a composite hydrogel could be predicted perfectly by using microstructural images. Using a Python script in the FE software, three datasets of microstructural images were generated with uniform, non-uniform, and irregular shapes of BGs that contained 3000 2D images per each dataset and their corresponding Young's moduli and Poisson's ratios. The datasets were also combined into a fourth dataset containing 9000 images. ResNet and AlexNet were implemented, tuned, and trained on the datasets. Different statistical metrics such as the MAE, MSE, and R-value were used to measure the performances of the implemented ResNet and AlexNet networks. The statistical descriptors showed that the ResNet and AlexNet networks had great performances in predicting the Young's modulus and Poisson's ratio whether the shape of the BGs distributed in the COL was circular or irregular. However, AlexNet had a relatively better performance with the ReLU activation function. This work could ease the extraction of composite hydrogels' mechanical properties and provide a surrogate model of FEM for the rapid prediction of the Young's modulus and Poisson's ratio of composite hydrogels, which could be further employed in developing a tool for tissue engineers and clinicians.

**Author Contributions:** Conceptualization, all authors (K.G., F.E. and R.B.); methodology, K.G.; software, K.G.; validation, all authors (K.G., F.E. and R.B.); investigation, K.G.; data curation, K.G.; writing—original draft preparation, K.G.; writing—review and editing, K.G. and F.E.; visualization, all authors (K.G., F.E. and R.B.); supervision, R.B. All authors have read and agreed to the published version of the manuscript.

**Funding:** Publication of this article was funded in part by the Open Access Subvention Fund and the John H. Evans Library.

**Data Availability Statement:** The training data for the study are available upon request.

**Conflicts of Interest:** The authors declare no conflict of interest.

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
