# Peer review of "Prediction of Composite Mechanical Properties: Integration of Deep Neural Network Methods and Finite Element Analysis"

_jcs, doi:10.3390/jcs7020054_

Round 1
Reviewer 1 Report
1. The title of the work is consistent with the content of the manuscript.
2. The analysis of the literature is appropriate and well-chosen. It gives a good background to the problem posed.
3. The description of the research methodology is somewhat incomplete. There is a lack of information regarding artificial neural networks. What they were and how they were tested to get the correct results. In addition, the results are very interesting and fairly well presented, the Komntarze could be more extensive.
4. Concluding conclusions are correctly formulated.
Reviewer 2 Report
please see the attachment

Round 2
Reviewer 1 Report
The additions made to the manuscript are sufficient. Explaining network architecture throws a new background to research. This allows for a full understanding of the set goal of the work. Of course, minor improvements could still be made.
Reviewer 2 Report
All major comments were adequately addressed and the Authors have done an admirable job of improving the quality of the manuscript. Therefore, it can be accepted without any structural modification.